# Molecular characterization of circulating *Salmonella* Typhi strains in an urban informal settlement in Kenya

Caroline Ochieng[1], Jessica C. Chen[2], Mike Powel Osita[1], Lee S. Katz[2], Taylor Griswold[2], Victor Omballa[1], Eric Ng'eno[1], Alice Ouma[1], Newton Wamola[1], Christine Opiyo[1], Loicer Achieng[1], Patrick K. Munywoki[3], Rene S. Hendriksen[4], Molly Freeman[2], Matthew Mikoleit[2], Bonventure Juma[3], Godfrey Bigogo[1], Eric Mintz[2], Jennifer R. Verani[2,3], Elizabeth Hunsperger[3], Heather A. Carleton[2]*

**1** Kenya Medical Research Institute, Centre for Global Health Research, Kisumu, Kenya, **2** Centers for Disease Control and Prevention, Atlanta, Georgia, United States of America, **3** Centers for Disease Control and Prevention-Kenya, Division of Global Health Protection, Nairobi, Kenya, **4** Technical University of Denmark, National Food Institute, DTU-Food. Kemitorvet, Denmark

* hcarleton@cdc.gov

**Data Availability Statement:** The raw sequence data have been submitted to the European Nucleotide Archive (http://www.ebi.ac.uk/ena) under accession no. ERP105715 or NCBI under the

## Abstract

A high burden of *Salmonella enterica* subspecies *enterica* serovar Typhi (*S.* Typhi) bacteremia has been reported from urban informal settlements in sub-Saharan Africa, yet little is known about the introduction of these strains to the region. Understanding regional differences in the predominant strains of *S.* Typhi can provide insight into the genomic epidemiology. We genetically characterized 310 *S.* Typhi isolates from typhoid fever surveillance conducted over a 12-year period (2007–2019) in Kibera, an urban informal settlement in Nairobi, Kenya, to assess the circulating strains, their antimicrobial resistance attributes, and how they relate to global *S.* Typhi isolates. Whole genome multi-locus sequence typing (wgMLST) identified 4 clades, with up to 303 pairwise allelic differences. The identified genotypes correlated with wgMLST clades. The predominant clade contained 290 (93.5%) isolates with a median of 14 allele differences (range 0–52) and consisted entirely of genotypes 4.3.1.1 and 4.3.1.2. Resistance determinants were identified exclusively in the predominant clade. Determinants associated with resistance to aminoglycosides were observed in 245 isolates (79.0%), sulphonamide in 243 isolates (78.4%), trimethoprim in 247 isolates (79.7%), tetracycline in 224 isolates (72.3%), chloramphenicol in 247 isolates (79.6%), β-lactams in 239 isolates (77.1%) and quinolones in 62 isolates (20.0%). Multidrug resistance (MDR) determinants (defined as determinants conferring resistance to ampicillin, chloramphenicol and cotrimoxazole) were found in 235 (75.8%) isolates. The prevalence of MDR associated genes was similar throughout the study period (2007–2012: 203, 76.3% vs 2013–2019: 32, 72.7%; Fisher's Exact Test: $P = 0.5478$, while the proportion of isolates harboring quinolone resistance determinants increased (2007–2012: 42, 15.8% and 2013–2019: 20, 45.5%; Fisher's Exact Test: $P < 0.0001$) following a decline in *S.* Typhi in Kibera. Some isolates (49, 15.8%) harbored both MDR and quinolone resistance determinants. There were no determinants associated with resistance to cephalosporins or azithromycin detected among the isolates sequenced in this study. Plasmid markers were only identified

BioProject PRJNA750407. The reference sequence assembly for isolate 2014K-0817 has been submitted to NCBI under the accession no AAOGUB000000000.

**Funding:** The funding for sequencing work was provided by the Bill and Melinda Gates Foundation Grant No. OPP1162700 through Henry Jackson Foundation Medical Research Institute Award number 65173 while PBIDS platform is supported by US CDC through a Cooperative Agreement #6U01GH002143-04-06 with Washington State University Global Health Kenya. The funders had no role in study design, data collection and analysis, decision to publish, or preparation of the manuscript.

**Competing interests:** The authors have declared that no competing interests exist.

in the main clade including IncHI1A and IncHI1B(R27) in 226 (72.9%) isolates, and IncQ1 in 238 (76.8%) isolates. Molecular clock analysis of global typhoid isolates and isolates from Kibera suggests that genotype 4.3.1 has been introduced multiple times in Kibera. Several genomes from Kibera formed a clade with genomes from Kenya, Malawi, South Africa, and Tanzania. The most recent common ancestor (MRCA) for these isolates was from around 1997. Another isolate from Kibera grouped with several isolates from Uganda, sharing a common ancestor from around 2009. In summary, *S.* Typhi in Kibera belong to four wgMLST clades one of which is frequently associated with MDR genes and this poses a challenge in treatment and control.

## Author summary

Typhoid fever is a major public health concern in endemic regions. Understanding the circulating strains of *S.* Typhi, could provide insight into the genomic epidemiology and guide in the choice of appropriate antibiotics. In this paper, our aim was to characterize *S.* Typhi strains causing invasive disease in Kibera, where a high typhoid burden has been described. We also aim to understand the evolutionary history of these strains and how antimicrobial resistance determinants have changed over time. We found that there was low diversity of *S.* Typhi observed in Kibera isolates with isolates grouping into 4 wgMLST clades and five genotypes. The majority (93.5%) of the isolates belonged to genotype 4.3.1; phylodynamic analysis suggest isolates of this genotype from Kibera are related to other 4.3.1 isolates from Africa and this genotype has been introduced multiple times in Kibera. This genotype in particular warrants close monitoring to inform antibiotic strategy in this population. Furthermore, concurrent detection of gene markers for MDR and quinolone resistance in some isolates raise concern about the potential emergence of extensive drug resistant (XDR) strains. Additional surveillance is needed in Kibera to monitor changing trends in resistance that may require altering clinical treatment, and to inform other preventive measures such as typhoid-conjugate vaccine introduction.

## Introduction

Typhoid fever is a systemic febrile illness caused by *Salmonella enterica* subspecies enterica serovar Typhi (hereafter referred to as *S.* Typhi). The global estimate of typhoid fever burden ranges between 11–21 million cases and approximately 128,000 to 161,000 deaths annually [1]. Increasing antimicrobial resistance (AMR) in *S.* Typhi complicates treatment and control of the disease in endemic regions. However, this increase is not uniform globally and has evolved at different rates in various endemic regions [2]. The first cases of *S.* Typhi isolates showing multidrug resistance (MDR), defined as co-occurring resistance to ampicillin, chlorampheni-col and co-trimoxazole, were reported in the early 1970s [3,4]. Later, ciprofloxacin resistance was also reported in majority of clinical isolates from endemic regions [5–7] and since late 2016 an extensive drug resistant (XDR) clone of *S.* Typhi with resistance to ceftriaxone has emerged and as a result of these changes some countries are shifting the recommended treatments to other classes of antimicrobial agents [8–11]. This evolving threat highlights the importance of monitoring circulating strains of *S.* Typhi for early detection of antimicrobial resistance patterns to guide on selection of effective antibiotics for patient management.

Whole-genome sequence (WGS)-based approaches using next-generation sequencing (NGS) have become effective tools to study genetic diversity and prediction of resistance phenotypes [12–18]. Extensive genomic studies of the *S*. Typhi strains are required to re-construct the full-scale evolutionary history and to understand the mutational events that have occurred over time [18]. Studies of the global population structure of *S*. Typhi have revealed a single clonal genotype, 4.3.1 (formerly described as haplotype 58 or H58) associated with MDR and increasing fluroquinolone resistance and date the emergence of this strain sometime in the mid to late 1980s or early 1990s, and indicate this strain has been increasing in population size since the early 1990s [17,19,20].

In Kibera, an urban informal settlement in Nairobi, Kenya, a high incidence (247 cases per 100,000 person-years of observation) of *S*. Typhi bacteremia was reported from 2007–2009 [21]. However, typhoid fever incidence can be dynamic over time [22], and declines in the Kibera typhoid fever incidence were observed from 2013 through 2017 [23]. Our objective was to characterize the *S*. Typhi strains causing invasive disease in Kibera over a 12-year period to understand the evolutionary history of these strains, their relationship to global typhoid isolates, and how antimicrobial resistance determinants have changed over time. To achieve these objectives, we sequenced genomes of invasive *Salmonella* isolates obtained from ongoing surveillance in Kibera.

## Methods

### Ethics statement

The population-based infectious disease surveillance (PBIDS) protocol for primary data collection was approved by Kenya Medical Research Institute's Scientific and Ethical Review Unit (SSC protocol number 1899 & 2761) and US Centers for Disease Control and Prevention (Protocol number 4566 and 6775). Written consent to participate in PBIDS was provided by heads of household at the time of enrollment. In addition, individual written informed consent from the patient (or parent/guardian) was obtained prior to sample collection.

### Study site

The study participants were residents of Kibera, an urban informal settlement in Nairobi, Kenya. An informal settlement in this context is an area where groups of temporary housing units have been constructed on land that the occupants have no legal claim. Kibera is characterized by high population density, limited access to safe water, and poor sanitation [21].

### Source of isolates

Isolates were derived from the Population-Based Infectious Disease Surveillance (PBIDS) platform, implemented by the Kenya Medical Research Institute in collaboration with the U.S. Centers for Disease Control and Prevention. PBIDS participants (~25,000 individuals) of all ages received free care for acute illness at a centrally located Tabitha Medical Clinic in Kibera run by Carolina for Kibera (CFK). A blood sample was collected from individuals presenting to the clinic who met severe acute respiratory illness or acute febrile illness case definitions as previously described [21]. Briefly, 8-10ml and 1-3ml of blood were collected (from persons ≥5 and children <5 years respectively) and inoculated in blood culture bottles then transported to Diagnostic and Laboratory System Program (DLSP) microbiology laboratory, a CDC-supported Kenya Medical Research Institute (KEMRI) laboratory in Kibera. BACTEC 9050 system alarm-positive bottles were sub-cultured using standard microbiology procedures [21]. Identified bacterial isolates were preserved in ultra-low freezers (-70C). For this study, all

*Salmonella* isolates that were available from blood culture collections from March 2007 –February 2019 were retrieved from the freezers. These were revived in Trypticase Soy agar (TSA-BD) media for 16–24 hours at 37˚C and DNA was extracted from all the viable isolates.

## DNA extraction and sequencing

The DNA sequencing of 412 Kibera isolates was conducted in three different institutions: Technical University of Denmark (DTU), Wellcome Sanger Institute, Cambridge, UK and KEMRI- DLSP laboratory in Nairobi. DNA extraction of 322 isolates was done using Wizard-Genomic DNA Purification Kit (Promega) and the rest by Qiagen DNeasy Blood & Tissue Kit (Qiagen) following the manufacturer's instructions. Genomic DNA of the 412 *Salmonella* isolates was used to create genomic libraries using the Nextera XT DNA sample preparation kit (Illumina Inc.) at DTU (n = 39), Sanger Institute (N = 322), and KEMRI-DLSP (n = 51). Following this procedure, the libraries were multiplexed, paired-end sequenced using Illumina platforms i.e., HiSeq 4000 by DTU, HiSeq X Ten by Wellcome Sanger Institute, and Miseq at KEMRI- DLSP. Raw sequence data from DTU and Wellcome Sanger Institute were transferred to KEMRI-DLSP for bioinformatics analysis. The raw sequence data have been submitted to the European Nucleotide Archive (http://www.ebi.ac.uk/ena) under accession no. ERP105715 or NCBI under the BioProject PRJNA750407. Accession numbers for individual sequences can be found in S1 Table.

## Data quality checks

General sequence data quality was checked using FastQC v0.11.15 tool [24]. Quality indicators of the sequence data were determined using SneakerNet v0.3 [25]. SneakerNet measures the average quality score of the forward and reverse reads and the combined genome coverage for each genome. We designated a coverage threshold of 30x and a minimum quality score of 30 for each read and if the q-score was below 30 an additional 10x coverage was required. Seq-Sero2 was used for WGS-based *Salmonella* serotyping (April 2019 alpha-test version) [26] and confirmation of laboratory culture serotyping. Contamination-free reads were determined by the absence of secondary genera in strains using MIDAS v. 1.3.0 [27] and Kraken 2 v. 2.0.8 [28], where the threshold for MIDAS is coverage $\geq 1.0x$ and for Kraken it is $\geq .5.0\%$. The absence of secondary *Salmonella* serotypes was monitored with SeqSero2 [26]. Isolate genomes that misidentified the species or serotype confirmation, identified the presence of secondary genera or secondary serotype above a respective threshold, or did not meet the required quality indicator thresholds were removed from downstream analyses.

## Sequence based subtyping

Whole genome multi-locus sequence typing (wgMLST) analysis was done using BioNumerics v. 7.6.3 (bioMérieux SA, Marcy-l'Étoile, France) [29]. An UPGMA tree was generated by determining the loci that were present in 95% of genomes (4177 loci) out of the total number of loci detected in the genomes (5082 loci). Sequence data was further analyzed using genotyphi v. 3 implemented in Pathogenwatch to determine *S.* Typhi genotype (https://github.com/katholt/genotyphi) [30,31].

   Resistance determinants and plasmid typing markers were identified using methods described by Tagg *et al.* 2020 [32]. Briefly, genomic sequence data were assembled *de novo* using shovill v. 1.0.9, with the–cov-cutoff set as 10% of the average genome coverage. Resulting assemblies were screened for resistance determinants using starAMR v. 0.4.0 using the databases from ResFinder (version updated on February 19, 2021) [33] (90% identity; 50% gene cutoff) and the PointFinder scheme for *Salmonella* (version updated on February 1, 2021).

Plasmid markers were identified using Abricate v.0.8.10 and a database adapted from Plasmid-Finder [34] (90% identity; 60% gene coverage). A fisher's exact test was performed to examine differences between AMR genotypes in two different study periods. Data were analyzed using the stats package for R version 4.1.1.

The wgMLST tree was annotated with resistance, plasmid, genotype, and year of isolation using iTOL v 6.4 [35].

### Phylogenetic analysis and molecular clock

To examine the relationship between *S*. Typhi in Kibera to global *S*. Typhi isolates and understand the emergence of *S*. Typhi in Kibera, we conducted a molecular clock analysis. For comparison, *S*. Typhi genomes from Wong *et al.* 2015 [17] and Park *et al.*2018 [37] were obtained from NCBI and characterized through the QC and subtyping methods outlined above. Additional isolates obtained from a study in Uganda were also included [38]. For each genotyphi-assigned genotype observed more than once in Kibera, a phylogeographic analysis was attempted. Due to the differences in sampling schemes within each country, up to 10 genomes from each country represented within a given genotype were sampled for inclusion in our phylogeographic dataset. We sampled to include a diversity of years, and sampled randomly within each year. For each genotype a separate hqSNP phylogeny was generated briefly as follows. Using Lyve-SET v1.1.4f [39] and the presets for *Salmonella enterica*, an alignment was generated using the sequence of 2014K-0817 as a reference (NCBI Accession: AAOGUB000000000). The resulting alignment was processed using Gubbins V.3.0.0 [40] to remove regions of the alignment having undergone recombination. Resulting phylogenies were analyzed using TempEST v 1.5.3. [41]. The best fitting root was selected and the correlation between root to tip divergence and time were examined using the correlation function. Genotype 4.3.1 displayed a moderate positive correlation and was selected for further analysis.

A discrete phylogeographic analysis was conducted using BEAST v. 2.6.4 based on the models which are part of the beast-classic 1.5.0 package [42], adding location as a discrete trait. The model averaging tool bModelTest v. 1.2.1 [43] was employed to select an appropriate substitution model for each genotype. To determine the tree model which best fits the data, for each genotype, all coalescent tree priors were evaluated (constant population, exponential population Bayesian skyline, and extended Bayesian skyline) using either a strict clock or a lognormal relaxed clock. Analysis was performed on the filtered SNP matrix generated using Lyve-SET and Gubbins as described in the previous section, and the xml file from Beauti was modified to account for constant sites (<data id = 'filt' spec = 'FilteredAlignment' filter = '-' data = '@filtOriginal' constantSiteWeights = '694048 783222 770094 689472'/>). All molecular clock analyses were run for 500,000,000 iterations, with sampling every 50,000 iterations with the first 10% of iterations discarded as burnin. Output was evaluated in Tracer v. 1.7.1 [44]. Three independent chains were run, and a representative BEAST tree file was selected for further processing. TreeAnnotator was used to produce a maximum clade credibility tree using the "median" options for heights. The maximum clade credibility tree was visualized using R v. 4.1.1 and the package ggtree [36].

## Results

### Genotypes of *S*. Typhi isolated in Kibera

Of the 412 Salmonella isolates from Kibera on which WGS was performed, 327 were characterized as *S*. Typhi on initial analysis. The remaining 85 isolates were identified as other non-typhoid serotypes and were excluded from further analysis. Of the 327 Typhi sequences, 17

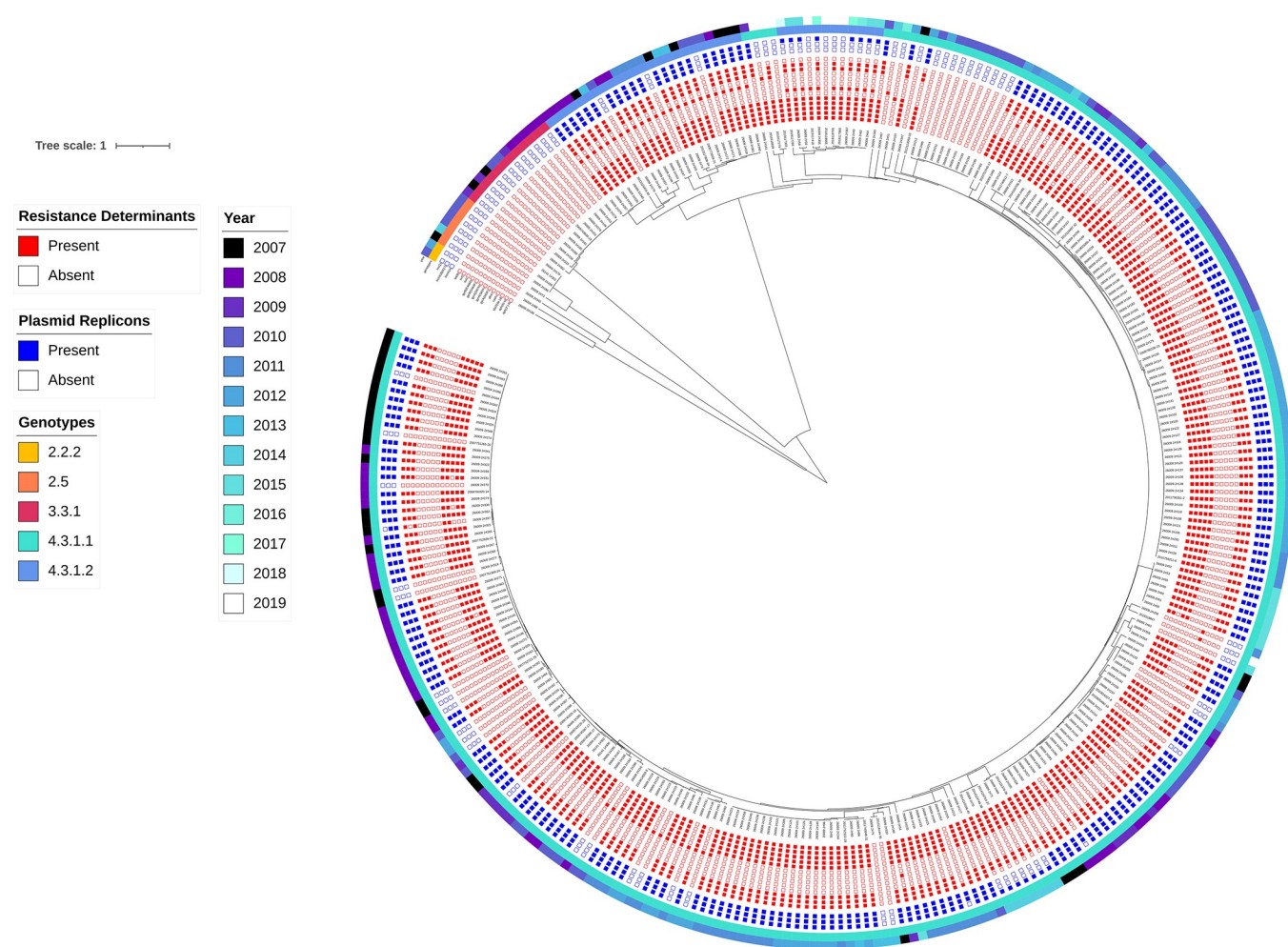

**Fig 1. wgMLST tree of 310 isolates of *S.* Typhi isolates collected in the Kibera settlement in Kenya colected from 2007–2019.** Displayed outside the tree (from inside to out) are the presence of antimicrobial resistance determiants (filled boxes in red), the presence of plasmid markers (filled blue boxes), genotyphi genotypes, and the year of isolation.

were dropped due to the presence of secondary genera or secondary serotype, or did not meet the required quality indicator threshold. Three hundred and ten isolates were identified as *S.* Typhi and were determined to have adequate sequence quality for further analyses (S1 Fig). The quality metrics of these sequences are available in S1 Table. By wgMLST, 4 different genetic clades were identified (Fig 1). The predominant clade captured most isolates detected from 2007 to 2019 (n = 290; 93.4%). Isolates in the predominant clade differed from each other by a median of 14 alleles (range: 0–52). All MDR isolates belonged to this clade and were genotype 4.3.1 which could be further segregated into 4.3.1.1 (n = 254, 81.9%) and 4.3.1.2 (n = 36, 11.6%). All 4.3.1.1 genomes belonged to the East Africa 1 sub-lineage (EA1), while 4.3.1.2 could be further sub-divided into EA2 (n = 24) and EA3 (n = 12). Isolates in the remaining 3 clades comprised of antimicrobial-susceptible isolates detected from 2007–2014 belonging to genotypes 2.2.2 (n = 2, 0.6%); 2.5 (n = 6, 1.9%); and 3.3.1 (n = 12, 3.9%). The two isolates belonging to genotype 2.2.2 formed a wgMLST clade and differed by 52 alleles. The six isolates in the genotype 2.5 clade displayed a median of 9.5 allele differences (range: 0–62), while the 12 isolates in the genotype 3.3.1 clade differed by a median of 0 alleles (range: 0–5).

## Genotypic characterization of antimicrobial resistance in the Kibera isolates

All the isolates with resistance determinants and plasmid replicons belonged to genotype 4.3.1.1 (lineage I) or 4.3.1.2 (lineage II) (Fig 1) with the former being more common than the latter. The other non-4.3.1 genotypes did not contain resistance markers or plasmids. We identified the following resistance determinants in the isolates: *aph* (3")-Ib and *aph*(6)-Id (confers aminoglycoside resistance) in 245 (79.0%) isolates; $bla_{TEM}$-1 (confers β-lactam resistance) in 239 (77.1%)isolates; *cat*A1 (confers chloramphenicol resistance) in 247 (79.6%) isolates and *dfr*A7 (confers trimethoprim resistance) in 247 (79.7%) isolates; *sul*1 and *sul*2 (confers sulphonamide resistance) in 243 (78.4%) and 240 (77.4%) isolates respectively and *tet*(B) (tetracycline resistance) in 224 (72.3%) isolates. The antibiotic resistance associated genes that contribute to MDR phenotype as described in this study include: $bla_{TEM}$-1, *cat*A1, *dfr*A7, *sul*1 and *sul*2. The prevalence of MDR genes was similar in earlier years where the number of *S.* Typhi identified was high (2007–2012: 203 isolates, 76.3%) and later years where *S.* Typhi levels were low (2013–2019: 32 isolates, 72.7%; Fishers Exact Test: $P$ = 0.5478) (Fig 2). Additionally, point mutations in the quinolone resistance-determining regions (QRDR) of *gyrA* or *gyrB* were detected in 62 (20.0%) isolates. These were: *gyrA*(S83F) in 19 (6.1%) isolates; *gyrA*(S83Y) in 12 (3.9%) isolates; *gyrA*(D87N) in 2 (0.6%) isolates; *gyrB*(S464F) in 27 (8.7%) isolates and *gyrB* (E466D) in 2 (0.6%) isolates. Of the isolates with mutations in the QRDR the majority also had acquired resistance (MDR) genes (n = 49/62; 79.0%) while only 13 isolates (21.0%) had only

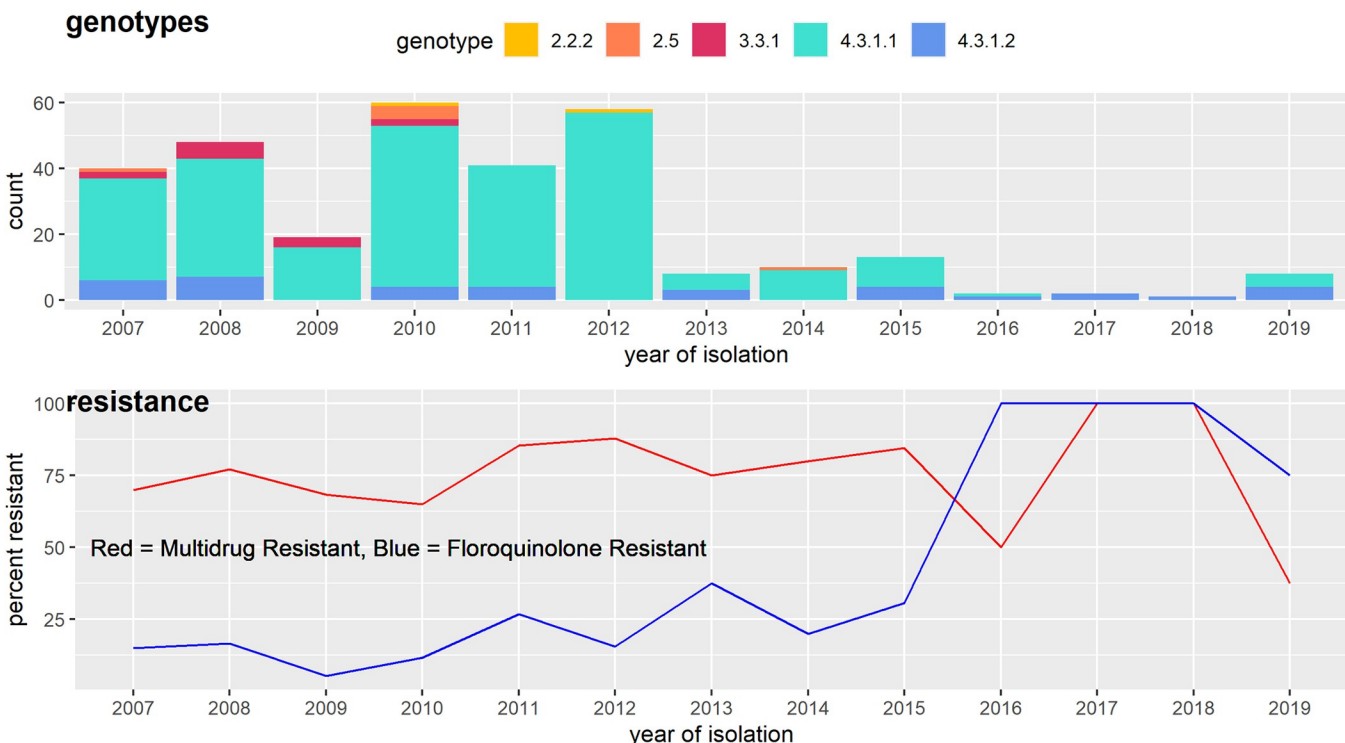

**Fig 2. Temporal trends in genotype and antibiotic resistance by year of isolation.** The top panel displays the genotype by year of isolation, where the x-axis is the year of isolation and the y-axis is count of isolates from that year belonging to each genotype. The bottom panel displays resistance information, where detection of determinants for a particular antimicrobial were employed as a proxy for resistance. The red line illustrates the percent of isolates over time with genes conferring resistance to ampicillin, chloramphenicol, and sulfa-methoxazole (defined as MDR). The blue line illustrates the percent of isolates with fluroquinolone resistance determinants.

QRDR mutations. Further, an increase in the proportion of isolates harboring quinolone resistance determinants was observed over the full study period (2007–2012: 42, 15.8% and. 2013–2019: 20, 45.5%; Fisher's Exact Test: $P<0.0001$) (Fig 2). No resistance determinants to 3rd and 4th generation cephalosporins nor carbapenems were detected as well as no mutations in the *acrB* gene or other determinants known to confer resistance to azithromycin.

Three different plasmid markers were also detected in the Typhi isolates including IncHI1A, IncHI1B(R27) and IncQ1. The majority of the isolates (241,77.7%) harbored one or more plasmid markers while 69(22.3%) had no plasmid markers at all. Of the 241 with plasmid markers, 223 (92.5%) had all the 3 markers, 15(6.2%) had IncQ1 only and 3(1.2%) had both IncHI1 markers only.

## Molecular clock analysis of genotype 4.3.1

A phylogeographic analysis of global genotype 4.3.1 isolates and isolates from Kibera was conducted to date the emergence of this genotype in Kibera (Fig 3). All isolates in this analysis (S2 Table) shared a most recent common ancestor (MRCA) dating back to approximately 1990 (median: 7/24/1990; 95% Highest Posterior Density (HPD) Interval: 7/9/1986–6/20/1992). The sampling of isolates from Kenya included eight isolates from Kibera, seven of which were genotype 4.3.1.1 EA1 and these isolates formed a clade with genomes from Kenya, Malawi, South Africa, and Tanzania. This clade had a MRCA dating back to around 1997 (median 8/20/1997; 95% HPD Interval 2/20/1995–9/11/2000). All but one of the Kibera isolates in this clade had resistance determinants and were MDR. The resistance *aph(3")-Ib*, *aph(6)-Id*, *bla*~TEM~, *catA1*, *dfrA7*, *sul1*, *sul2*, and *tet(B)* were detected in MDR isolates, and one isolate had an additional *gyrA*(S83F) mutation. The remaining Kibera isolate was genotype 4.3.1.2 EA3 and grouped with isolates from Uganda. These isolates share a common ancestor from around 2009 (median 6/22/2009: 95% HPD Interval 7/7/2003–1/29/2013). The Kibera isolate in this clade had the following resistance determinants, *aph(3")-Ib*, *aph(6)-Id*, *bla*~TEM~, *catA1*, *dfrA7*, and a *gyrA*(S83Y) mutation. An additional genome from Kenya from a previous study [17] grouped with isolates from India and was genotype 4.3.1.2. EA2. This analysis highlights multiple distinct introductions of the 4.3.1genotype in Kibera.

## Discussion

The genomic data from our study provide insight into the *S*. Typhi population that has been causing invasive disease in Kibera for more than a decade. We also identified 4 different genetic clades amongst the Kibera isolates with the dominant clade persisting throughout the study period. The predominant clade comprised of *S*. Typhi genotypes 4.3.1.1 and 4.3.1.2 and frequently harbored IncHI1 plasmids, which have been reported to contribute to the success of dominant MDR *S*. Typhi haplotypes [45]. This might help explain the persistence of this genotype in Kibera throughout the study period. The other three genotypes lacked resistance determinants and plasmid markers and were isolated only infrequently throughout the study period and were not detected after 2014 because they could have been displaced by the dominant strain.

Further analysis showed that, all the isolates with resistance markers belonged to *S*. Typhi genotype 4.3.1. The majority of these isolates had MDR genes and the percentage of MDR remained relatively consistent over time. However, the transmission of the ESBL and azithromycin producing *S*.Typhi has not yet spread to the African continent and surveillance for this should be strengthened. The MDR genes were associated with IncHI1 plasmids which are known carriers of MDR genes and are also associated with the H58 Typhi haplotype, now denoted as genotype 4.3.1. The 69 (22.3%) MDR isolates without plasmids could have had the

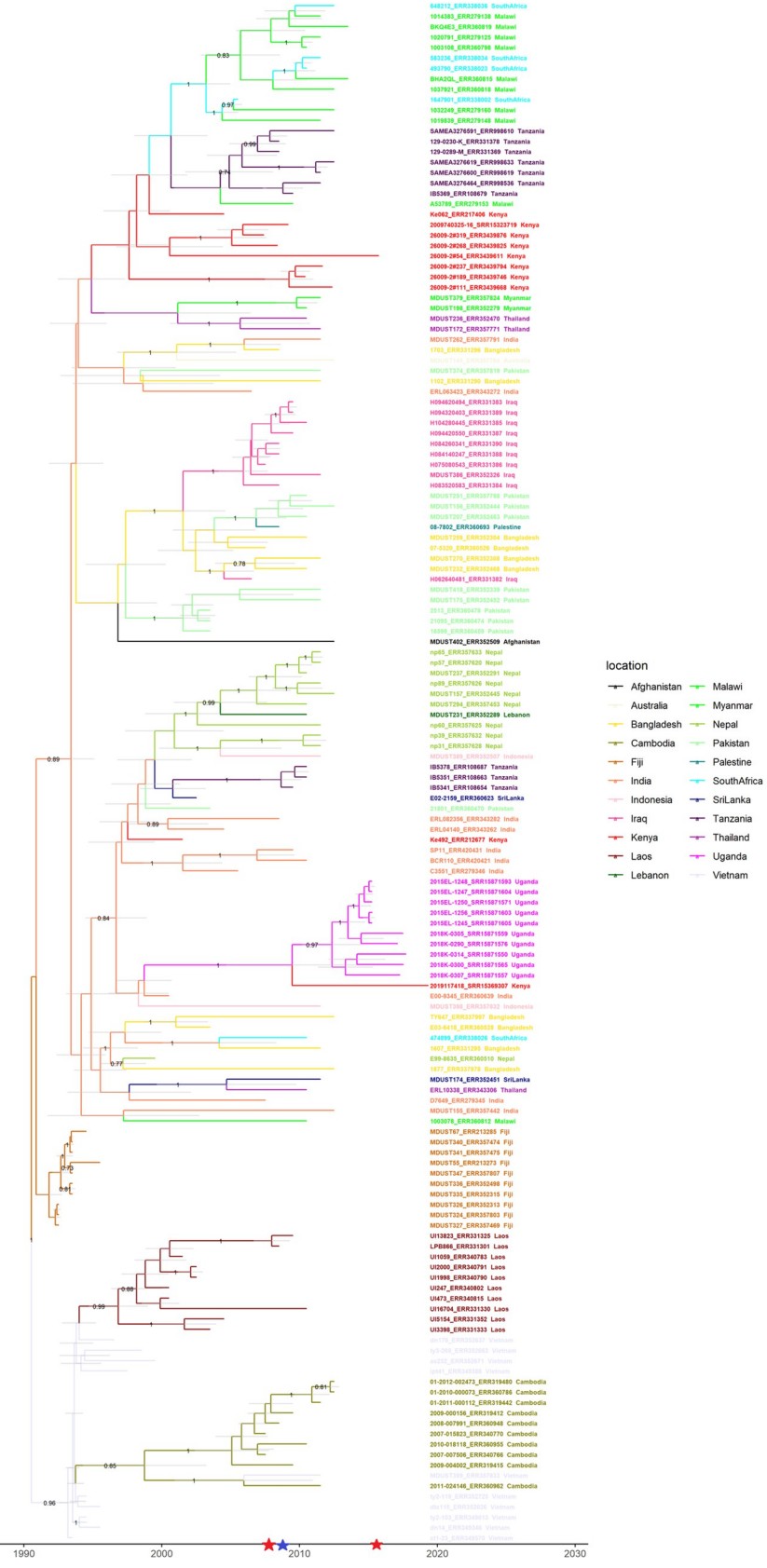

**Fig 3. Global context of 4.3.1 genomes from Kibera using a molecular clock analysis.** Tip dated maximum clade credibility tree of 148 isolates of *S.* Typhi genotype 4.3.1 generated using BEAST2. This analysis includes 8 isolates from the Kibera settlement in Kenya sequenced as part of the present study, 10 isolates from Uganda, and the remaining isolates are from two previous studies of the phylogeography of *S.* Typhi published by Wong et al (2015) and Park et al. (2018). The x-axis denotes calendar year. The tree is colored by location, with isolates from Kenya highlighted in red. Tip labels include the isolate ID and accession ID separated by an underscore. The country from which the sample was isolated from is also displayed. Posterior support for internal nodes are displayed where values are >0.70. Grey horizontal bars indicate the 95% Highest Posterior Density (HPD) Interval for height of the given clade (corresponds to age). Colored star markers on the x-axis indicate relevant epidemiological events. Red stars indicate typhoid outbreaks in Uganda, and the blue star marks when improvements in water sanitation were made in Kibera [50].

MDR genes integrated into their chromosomes thereby losing the plasmids in the process [17]. We also observed chromosomal point mutations on DNA gyrase subunits A and B but none on topoisomerase IV gene. While a previous study of global *S.*Typhi strains showed that African strains, including those from Kenya, had increased MDR but no *gyrA* mutations [17], we report a combination of both in some isolates. A greater proportion of the isolates with mutations on QRDR (*gyrA/B*) also had MDR genes which increase the possibility XDR strains could emerge. The increase in fluoroquinolone resistance determinants in 2016–2019 isolates could have been caused by indiscriminate use of fluoroquinolones (e.g. ciprofloxacin) for a period of time [17]. The issue of over- the- counter drugs and incompletion of dosage are also common practices in Kibera and could have contributed too. Notably, we did not detect transmissible fluoroquinolone resistance markers in the Kibera isolates within the study period.

Phylogeographic analysis of isolates from Kibera along with additional global *S.* Typhi isolates [17,37] suggests that multiple introductions of genotype 4.3.1 occurred in Kibera, and are consistent with the EA1-3 4.3.1 sub-lineages circulating in Kenya [46]. The MRCA of isolates in this analysis was similar to that previously reported thus confirming our approach [17]. The majority of the Kibera isolates belonged to 4.3.1.1 EA1 and show a close genetic and temporal relationship with other isolates from Africa, specifically from Malawi, South Africa, and Tanzania. Clustering of isolates from these countries was initially reported by Wong *et al.* 2015 and we estimate these isolates share a common ancestor from around 1997 [17]. Kariuki *et al*, estimate the emergence of the EA1 sub-lineage to be around 1990 [46]. Six out of seven of the genomes from Kibera possessed resistance genes commonly reported in the H58 isolates (*aph (3")-Ib*, *aph(6)-Id*, *bla*$_{TEM}$, *catA1*, *dfrA7*, *sul1*, *sul2*, and *tet(B)*), and one of these isolates contained an additional *gyrA(*S83F) mutation previously reported to be present in 45% of H58 isolates, and rare in EA1 [17,47], while the remaining isolate did not have any resistance determinants.

The remaining isolate from Kibera in this analysis was isolated in 2019 and belonged to 4.3.1.2 EA3 and shows a close genetic and temporal relationship with isolates from Uganda collected in 2015 and 2018. The MRCA of this clade dates back to approximately 2009, after which several large outbreaks have been reported in Uganda [48,49], and Kariuki *et al*. further date the emergence of EA3 in Kenya to be around 2012 [46]. The 2019 isolate also possessed resistance genes common to H58 (*aph(3")-Ib*, *aph(6)-Id*, *bla*$_{TEM}$, *catA1*, *dfrA7*) in addition to the *gyrA*(S83Y) mutation previously reported to be present in 9% of H58 isolates overall, but conserved among EA3 [17,46]. Overall these data indicate multiple introductions of MDR 4.3.1 (H58) into Kenya and continued monitoring may help better elucidate pathways of spread in the region and help identify control measures.

We found several limitations in our study. One of the limitations of this study is that data are from an urban informal settlement which may not be representative of rural settlements or other urban areas in Kenya. This study utilizes short-read sequencing which provides valuable

information to perform genomic characterization, however complete assembly of plasmids is challenging with this technology due to insertion sequence elements and other repeat elements in the plasmid sequence. Additional study is required to associate specific resistance genes with specific plasmids, as well as facilitate comparison with known reference plasmids previously identified in *S*. Typhi. Limitations of the molecular clock analysis include the detection of only a moderate temporal signal, which weakens our ability for more precise estimation of divergence events. A moderate temporal signal was also observed by Wong et al. which they attribute to sampling of isolates over a short time frame [17]. Differences in sampling schemes in different regions may influence molecular clock results; however we attempted to mitigate this by subsampling data to not allow for too many sequences from a particular country. Regional data gaps may also exist, which may challenge our interpretation of the global evolutionary history of *S*. Typhi.

## Conclusion and recommendation

Low divergence of S. Typhi was observed in Kibera isolates with isolates grouping into 4 wgMLST clades and five genotypes, of which one clade comprised of genotypes 4.3.1.1 and 4.3.1.2 and contained the majority of isolates. The presence of MDR genotype 4.3.1 in this population is of clinical and public health importance and warrants monitoring to guide empiric antibiotic treatment in this context. Additionally, the coexistence of MDR gene markers with fluoroquinolone resistance markers in the Kibera isolates reflects the potential for emergence of extensively drug resistant (XDR) strains in this population. The transmission of the ESBL and azithromycin producing S.Typhi has not yet spread to the African continent and surveillance for this should be strengthened to monitor changing trends in resistance that may require altering in clinical treatment and additional preventive measures such as TCV vaccine introduction decisions.

## Disclaimer

The findings and conclusions in this report are those of the author and do not necessarily represent the official position of the Centers for Disease Control and Prevention. Use of trade names is for identification only and does not imply endorsement by the Centers for Disease Control and Prevention or by the U.S. Department of Health and Human Services.

## Supporting information

**S1 Fig. Flow chart of isolates used in this study (March 2007-February 2019).**
(TIF)

**S1 Table. Accession numbers, metadata and sequence data characteristics for isolate sequence data generated for this study.**
(XLSX)

**S2 Table. Genome identifiers for genotype 4.3.1 molecular clock analysis.**
(XLSX)

## Acknowledgments

We wish to acknowledge the Wellcome Sanger Institute for their assistance with sequencing of majority of the isolates. We thank the field and laboratory teams who conducted the surveillance that generated the isolates, and the PBIDS population for their willingness to participate in the surveillance study. We also acknowledge Ana Lauer for contributing the Uganda

sequences to the study. We acknowledge Geoffrey Masyongo for coordinating field activities and data collection.

## Author Contributions

**Conceptualization:** Caroline Ochieng, Molly Freeman, Matthew Mikoleit, Bonventure Juma, Eric Mintz, Jennifer R. Verani, Elizabeth Hunsperger, Heather A. Carleton.

**Data curation:** Caroline Ochieng, Jessica C. Chen, Mike Powel Osita, Taylor Griswold, Heather A. Carleton.

**Formal analysis:** Caroline Ochieng, Jessica C. Chen, Taylor Griswold, Heather A. Carleton.

**Funding acquisition:** Eric Mintz, Jennifer R. Verani, Elizabeth Hunsperger, Heather A. Carleton.

**Investigation:** Caroline Ochieng.

**Methodology:** Caroline Ochieng, Jessica C. Chen, Mike Powel Osita, Lee S. Katz, Taylor Griswold, Victor Omballa, Eric. Ng'eno, Alice Ouma, Newton Wamola, Christine Opiyo, Loicer Achieng, Patrick K. Munywoki, Rene S. Hendriksen, Molly Freeman, Matthew Mikoleit, Bonventure Juma, Godfrey Bigogo, Heather A. Carleton.

**Project administration:** Caroline Ochieng, Elizabeth Hunsperger, Heather A. Carleton.

**Resources:** Rene S. Hendriksen.

**Software:** Jessica C. Chen, Lee S. Katz.

**Supervision:** Jennifer R. Verani, Elizabeth Hunsperger, Heather A. Carleton.

**Visualization:** Jessica C. Chen, Heather A. Carleton.

**Writing – original draft:** Caroline Ochieng, Jessica C. Chen, Jennifer R. Verani, Elizabeth Hunsperger, Heather A. Carleton.

**Writing – review & editing:** Caroline Ochieng, Jessica C. Chen, Mike Powel Osita, Lee S. Katz, Taylor Griswold, Victor Omballa, Eric. Ng'eno, Alice Ouma, Newton Wamola, Christine Opiyo, Loicer Achieng, Patrick K. Munywoki, Rene S. Hendriksen, Molly Freeman, Matthew Mikoleit, Bonventure Juma, Godfrey Bigogo, Eric Mintz, Jennifer R. Verani, Elizabeth Hunsperger, Heather A. Carleton.

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
