## [Decision Letter · Decision Letter 0]

18 Jul 2022

Dear Dr. Carleton,

Thank you very much for submitting your manuscript "Molecular characterization of circulating Salmonella Typhi strains in an urban informal settlement in Kenya" for consideration at PLOS Neglected Tropical Diseases. As with all papers reviewed by the journal, your manuscript was reviewed by members of the editorial board and by several independent reviewers. The reviewers appreciated the attention to an important topic. Based on the reviews, we are likely to accept this manuscript for publication, providing that you modify the manuscript according to the review recommendations. 

In your revised manuscript, please address the comments of Reviewer 2 regarding whether the antibiotic resistance markers are linked to the specific plasmids whose markers were identified in the study.

Sincerely,

Travis J Bourret

Academic Editor

Alfredo Torres

Section Editor

In your revised manuscript, please address the comments of Reviewer 2 regarding whether the antibiotic resistance markers are linked to the specific plasmids whose markers were identified in the study.

Reviewer's Responses to Questions

**Key Review Criteria Required for Acceptance?**

**Methods**

-Are the objectives of the study clearly articulated with a clear testable hypothesis stated?

-Is the study design appropriate to address the stated objectives?

-Is the population clearly described and appropriate for the hypothesis being tested?

-Is the sample size sufficient to ensure adequate power to address the hypothesis being tested?

-Were correct statistical analysis used to support conclusions?

-Are there concerns about ethical or regulatory requirements being met?

Reviewer #1: Objective of the study is clearly mentioned. Hypothesis and introduction is well written

Reviewer #2: This study entitled "Molecular characterization of circulating Salmonella Typhi strains in an urban informal

settlement in Kenya" was carried out to determine the evolution and relationship of local strains from ongoing surveillance from Kenya to the global Salmonella lineages, and to determine the changes occurred for antimicrobial resistance over time in the examined isolates.

The following points should be addressed:

- Why authors used the ggtree package if it has the same function like ITOL for the annotation of phylogenomic trees.

-As mentioned in Paragraph (lines 262-266), three different plasmid markers were detected among isolates; the authors have not determined if any of the resistance genes detected are linked to a specific plasmid replicon type, if so, it is important to reconstruct the plasmid sequences from whole genome sequences.

**Results**

-Does the analysis presented match the analysis plan?

-Are the results clearly and completely presented?

-Are the figures (Tables, Images) of sufficient quality for clarity?

Reviewer #1: Results are well presented

Reviewer #2: As mentioned in Paragraph (lines 262-266), three different plasmid markers were detected among isolates; the authors have not determined if any of the resistance genes detected are linked to a specific plasmid replicon type, if so, it is important to reconstruct the plasmid sequences from whole genome sequences.

**Conclusions**

-Are the conclusions supported by the data presented?

-Are the limitations of analysis clearly described?

-Do the authors discuss how these data can be helpful to advance our understanding of the topic under study?

-Is public health relevance addressed?

Reviewer #1: Conclusion is based on result

Reviewer #2: (No Response)

**Editorial and Data Presentation Modifications?**

Reviewer #1: Source of Salmonella isolates clearly indicated. Molecular typing methods are standard.

Reviewer #2: (No Response)

**Summary and General Comments**

Reviewer #1: Submitted manuscript as in its present form is of publishable standard and provides new insights on circulating genotypes in Kibera

Reviewer #2: (No Response)

PLOS authors have the option to publish the peer review history of their article (what does this mean?). If published, this will include your full peer review and any attached files.

Reviewer #1: Yes: *Prof.Dr.Dwij Raj Bhatta, PhD microbiology

Reviewer #2: No

Figure Files:

Data Requirements:

Reproducibility:

References

---

## [Editor Report · Decision Letter 1]

28 Jul 2022

Dear Dr. Carleton,

We are pleased to inform you that your manuscript 'Molecular characterization of circulating Salmonella Typhi strains in an urban informal settlement in Kenya' has been provisionally accepted for publication in PLOS Neglected Tropical Diseases.

Best regards,

Travis J Bourret

Academic Editor

Alfredo Torres

Section Editor

---

## [Editor Report · Acceptance letter]

18 Aug 2022

Dear Dr. Carleton,

We are delighted to inform you that your manuscript, "Molecular characterization of circulating Salmonella Typhi strains in an urban informal settlement in Kenya," has been formally accepted for publication in PLOS Neglected Tropical Diseases.

Best regards,

Shaden Kamhawi

co-Editor-in-Chief

Paul Brindley

co-Editor-in-Chief
